# VideoVLA: Video Generators Can Be Generalizable Robot Manipulators

**Yichao Shen**[1,2*], **Fangyun Wei**[2†], **Zhiying Du**[3*], **Yaobo Liang**[2], **Yan Lu**[2],
**Jiaolong Yang**[2†], **Nanning Zheng**[1†], **Baining Guo**[2]

[1]IAIR, Xi'an Jiaotong University[§]    [2]Microsoft Research Asia    [3]Fudan University

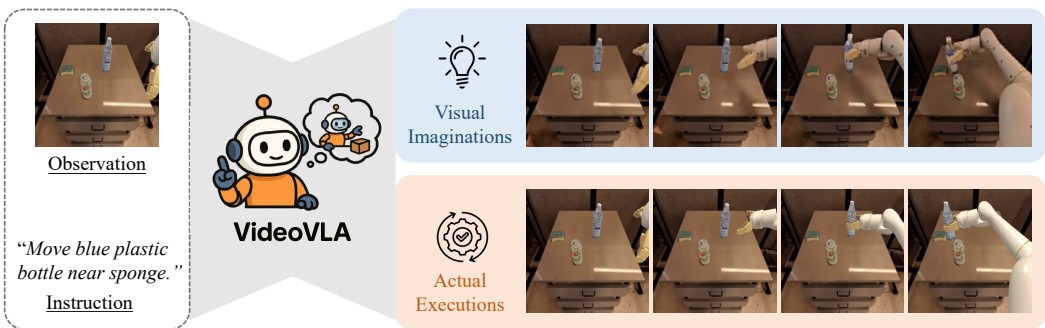

Figure 1: Illustration of VideoVLA. Given a language instruction and the current visual observation, VideoVLA jointly predicts the appropriate sequence of next actions and generates video content that illustrates how these actions will influence physical interactions in the environment. In addition to delivering strong performance on in-domain tasks, VideoVLA demonstrates robust generalization to novel objects and unseen skills. This capability stems from its use of pre-trained *video generation models*—distinct from prior vision-language-action approaches [1, 2, 3, 4, 5, 6, 7] that primarily rely on pre-trained vision-language *understanding models*—as well as its dual-objective strategy.

## Abstract

Generalization in robot manipulation is essential for deploying robots in open-world environments and advancing toward artificial general intelligence. While recent vision-language-action models leverage large pre-trained understanding models for perception and instruction following, their ability to generalize to novel tasks, objects, and settings remains limited. In this work, we present VideoVLA, a simple approach that explores the potential of *directly transforming large video generation models into robotic VLA manipulators*. Given a language instruction and an image, VideoVLA predicts an action sequence as well as the future visual outcomes. Built on a multi-modal Diffusion Transformer, VideoVLA jointly models video, language, and action modalities, using pre-trained video generative models for joint visual and action forecasting. Our experiments show that high-quality imagined futures correlate with reliable action predictions and task success, highlighting the importance of visual imagination in manipulation. VideoVLA demonstrates strong generalization, including imitating other embodiments' skills and handling novel objects. This dual-prediction strategy—forecasting both actions and their visual consequences—explores a paradigm shift in robot learning and unlocks generalization capabilities in manipulation systems.

---

[*]Microsoft Research Interns.    [†]Corresponding Authors.

[§]National Key Laboratory of Human-Machine Hybrid Augmented Intelligence, National Engineering Research Center for Visual Informationand Applications, and Institute of Artificial Intelligence and Robotics, Xi'an Jiaotong University

39th Conference on Neural Information Processing Systems (NeurIPS 2025).

# 1 Introduction

Generalization has long been a central goal in robot manipulation, representing a critical step toward achieving artificial general intelligence. The aspiration is for robots not only to perform tasks encountered during training but also to handle unseen tasks, manipulate novel objects, and operate in unfamiliar environments. This capability is critical for deploying robots in open-world settings, where unpredictability and variability are the norm rather than the exception. Recently, researchers have begun investigating whether robotic systems exhibit an emergence point—where generalization capabilities begin to surface—by scaling both the volume of training data and the size of model parameters, analogous to the scaling laws observed in Large Language Models (LLMs) [8, 9]. This line of inquiry has led to the development of Vision-Language-Action (VLA) models [1, 3, 4, 10]. Unlike conventional robot manipulation architectures [11, 12], VLA models typically incorporate billions of parameters, perceive complex visual environments, follow natural language instructions, and, crucially, leverage large pre-trained vision-language [13, 14, 15], vision [16, 17, 18], or language [19, 20, 21] models. These pre-trained models serve two purposes: (1) reducing the need for extensive task-specific robotic data, and (2) enabling the transfer of knowledge from general-purpose models to robotic manipulation tasks. Existing works [2, 3, 4, 10, 22] have demonstrated that using pre-trained vision-language understanding models as the VLA model backbone and finetuning them on robot action datasets can significantly reduce data requirements and achieve remarkable task performance. However, despite these advances, true generalization—particularly to unseen tasks, objects, or environments—remains limited and is yet to be fully realized.

Recently, large video generation models [23, 24, 25, 26] have demonstrated remarkable generalization capabilities when conditioned on novel textual or visual inputs. The generated videos exhibit exceptional outstanding physical plausibility [25, 26], which can be attributed to the models' extensive knowledge learned from massive real-world videos. Notably, the situation of video generators handling novel text and novel image conditions shows a natural alignment with that of robot manipulators dealing with unseen instructions and unseen observations. The understanding of physical dynamics learned by video generators is also a fundamental capability required for any high-performing robotic manipulator to reason about the physical consequences of their generated actions. Furthermore, video generators can predict future world states by following given instructions, which inherently reflects a planning capability that is also crucial for robotic manipulation models to anticipate and organize their interactions with the physical environment. Motivated by these observations, we aim to explore the following question: ***"Can large video generators be seamlessly adapted into generalizable robotic manipulators?"***

To answer this question, this work studies constructing and fine-tuning VLA models based on large video generation models. To bridge the gap between video generation and robotic manipulation, the key lies in enabling video generation models to produce instruction-following actions that can be executed by robots. Furthermore, to effectively transfer the strong generalization abilities of video generation models in the video domain to the action domain, it is essential to ensure a consistency between the actual execution of generated actions and the visual imagination represented in the generated videos (see Figure 1 for as illustration). This alignment allows the semantic and physical coherence learned in video generation to be naturally extended to embodied robotic behaviors.

We propose a simple yet effective approach called VideoVLA, which transforms a Video Diffusion Transformer [24] to a Video-Action Diffusion Transformer by adding actions as a new output modality and jointly denoising video and action. The Diffusion Transformer (DiT) [27] architecture has demonstrated remarkable superiority in both video generation [24, 28, 26] and action generation [4, 10, 7] tasks, and in this work we unify them into a single, multi-modal DiT framework. In this multi-modal DiT, the video and language are encoded using specific tokenizers—such as a causal video VAE [24] and a T5 [29] text encoder—following standard practices in generative modeling. We do not apply tokenization to the action modality and simply use the 7-D vector representation encoding robot translation, rotation, and gripper state. The model operates by taking the language tokens and the latent of the current visual observation as conditions; it then jointly predicts the future actions and generates the corresponding future visual contents that would result from executing these ions in the current environment, supervised by a DDPM Diffusion loss [30].

Our experiments reveal a strong correlation between the predicted actions and the generated video clips—when the imagined future (i.e., the generated videos) closely aligns with the actual outcome of the environment, the corresponding predicted actions consistently yield a higher task success rate. This observation suggests that the quality of visual imagination serves as an implicit indicator of

action reliability. In other words, when VideoVLA produces coherent and plausible future visual predictions, it is more likely that the associated actions are accurate for task completion. This finding underscores the importance of jointly predicting future visual dynamics alongside future actions.

One of the key advantages of VideoVLA lies in its generalization capability. Beyond performing in-domain tasks and manipulating objects seen during training, it demonstrates promising generalization in two challenging settings: (1) imitating skills from other embodiments, and (2) handling novel objects that do not appear in the training set. This capability can be attributed to two factors: (1) the use of pre-trained video generation models, which enable the system to interpret language instructions and generate plausible imagined futures, and (2) VideoVLA's dual-prediction strategy, which fosters a strong correlation between predicted actions and their corresponding visual consequences. VideoVLA explores a paradigm shift by pioneering the use of pre-trained video generation models for robot manipulation, in contrast to the dominant reliance on pre-trained understanding models in prior VLA works. The feasibility and potential of such a new paradigm are clearly demonstrated by this work. As generative models continue to improve, we believe that robotic manipulation systems derived from video generation models, like VideoVLA, will exhibit increasingly robust generalization, moving us closer to the goal of artificial general intelligence.

## 2 Related Works

**Vision-Language-Action Models.** Recent advances in vision-language-action (VLA) models [1, 2, 3, 4, 10, 5, 6, 7] have enabled a new paradigm for language-guided robot manipulation—these models interpret language instructions, perceive the current environment, and control the robot to complete tasks in an end-to-end manner. This success is largely driven by the advancement of vision [16, 17, 18], language [19, 20, 21], and vision-language [13, 14, 15] foundation models. By leveraging these pre-trained understanding-oriented models, VLA systems have achieved notable improvements in task success rates and demonstrated a certain degree of generalization to novel objects [3, 31] and unfamiliar environments [4, 6]. Another key driver in the advancement of VLA models is the increasing availability of large-scale, diverse datasets. The RT series [32, 33] demonstrates that VLA models can effectively leverage large-scale datasets and deliver promising results. To support continued progress, the Open X-Embodiment [2] dataset was released, aggregating over 60 robot datasets—including Language Table [34], Berkeley Bridge [35], and others—into a unified benchmark for pre-training. VideoVLA builds upon the rapid progress in robot datasets and pre-trained foundation models but departs from prior VLA approaches that rely primarily on understanding-oriented models. Instead, VideoVLA leverages large-scale pre-trained video generation models and introduces a dual-prediction strategy—jointly predicting future actions and corresponding visual imaginations that depict the anticipated outcomes of those actions. Recent works, UVA [36] and VPP [37], represents initial steps toward this paradigm. Compared to these efforts, VideoVLA differs in four key aspects beyond architectural distinctions: (1) we fully exploit the capabilities of large-scale pre-trained video generators; (2) our model demonstrates strong generalization to novel objects and unseen skills; (3) we reveal a strong correlation between predicted actions and visual imaginations; and (4) VideoVLA achieves competitive performance against recent state-of-the-art VLA models such as $\pi_0$ [10] and CogACT [4] in both simulated and real-world settings.

**Video Generation Models.** Video generation [38, 23, 24, 25] aims to synthesize realistic and temporally coherent video sequences. This field has experienced rapid progress, largely fueled by advances in image generation [39] and the development of diffusion models [27, 40]. Modern video generation frameworks typically support multi-modal conditioning. For example, text-to-video models [41, 42, 38] generate videos from natural language prompts, while image-to-video and pose-to-video models animate static inputs using learned motion priors [43, 44]. Most state-of-the-art video generation methods adopt a two-stage pipeline: first, a video VAE [24, 25] is trained to encode each video into a sequence of latent representations; then, a diffusion transformer such as DiT [27] or MM-DiT [39] is trained with a diffusion loss to model the spatiotemporal distribution over these video latents. Our model is built upon CogVideoX [24]—one of the most powerful video generation models to date. To our knowledge, this is the first work that adapts a large-scale video generator to the domain of robotic manipulation, demonstrating that such models can serve not only generative purposes but also as effective tools for visual planning and action prediction.

**Video Generation for Robot Manipulation.** Using video generation models for robot manipulation has been an active area of research in recent years. However, most existing approaches incorporate video generation as a visual planning component within modular frameworks to assist with action prediction. For instance, [45, 46, 47] extract actions from video predictions using inverse dynamics

models. [48] estimates end-effector actions by computing optical flow between frames and leveraging depth maps. Approaches like [49, 50] extract a goal frame from the predicted video and use it as the condition for action prediction. [51] and [52] focus on human hand-object interaction video generation and design models for human-to-robot transfer. [53] gathers future frame information by querying features from video generation model and uses a diffusion head to generate action. [54] and [6] generate video frames autoregressively and predict one frame at a time along with an action. Different from these works, our method introduces a unified, end-to-end VLA model directly adapted from a large pre-trained video generation model, which jointly predicts the video and action modalities within a single large transformer architecture, and it leverages the generalization capabilities learned from video generation pretraining on massive real-world videos.

## 3 Methodology

### 3.1 Problem Formulation

Given a text instruction $\mathcal{T}$ describing a task and the current visual observation $\mathcal{O}$, the objective is to jointly predict:

1. An action chunk $\mathcal{A} = \{a_i \in \mathbb{R}^7\}_{i=1}^K$ consisting of $K$ actions to be executed, where each action $a_i$ is a 7-D vector. The first three dimensions encode the wrist rotation, the next three dimensions encode the wrist translation, and the final dimension indicates the gripper state—a binary value, where 0 denotes a closed gripper and 1 denotes an open gripper.

2. A video clip $\mathcal{F} = \{F_j\}_{j=1}^N$ composed of $N$ frames that depict the anticipated future visual content resulting from executing $\mathcal{A}$. In our implementation, we do not predict the raw visual frames directly; instead, we predict their latent representations, as detailed in Section 3.3.

After executing the predicted action chunk $\mathcal{A}$, a new observation is obtained, and the process is repeated—that is, the model predicts the next action chunk based on the updated observation—until the task is completed. Note that the frequencies of $\mathcal{A}$ and $\mathcal{F}$ may differ—each action $a_i$ in $\mathcal{A}$ can correspond to multiple frames in $\mathcal{F}$.

### 3.2 Overview

Figure 2 presents an overview of VideoVLA, which primarily consists of: (1) two encoders that transform each input modality—the language instruction and the video clip—into tokens or latent representations; and (2) a DiT backbone that, conditioned on the language tokens and the latent representation of the current visual observation, jointly predicts future frame latents and the corresponding action sequence necessary to accomplish the task.

### 3.3 Data Preprocessing

VideoVLA processes language and visual inputs in the latent space to improve computational efficiency. To achieve this, it employs a text encoder and a video encoder, as shown in Figure 2(a).

**Text Encoder.** We adopt T5 [29] as our text encoder to convert each language instruction $\mathcal{T}$ into a fixed-length token sequence of 226 tokens. The resulting encoded tokens are denoted as $T$.

**Video Encoder.** The video encoder aims to transform a video clip $\mathcal{F} = \{F_j\}_{j=1}^N$, consisting of $N$ frames, into a sequence of video latents $\mathcal{V} = \{V_j \in \mathbb{R}^{h \times w}\}_{j=1}^n$, where $n$ is the number of video latents after temporal downsampling, and $(h, w)$ represents the spatial resolution of each latent $V_j$.

We adopt the 3D-causal VAE encoder from CogVideoX [24] as our video encoder. Owing to its causal design, the first video latent $V_1$ in the generated sequence $\mathcal{V} = \{V_j\}_{j=1}^n$ encodes only the first frame $F_1$, which corresponds to the current visual observation $\mathcal{O}$. In other words, $V_1$ serves as the latent representation of the current observation. During inference, we encode the current visual observation $\mathcal{O}$ alone to obtain $V_1$, whereas during training, the entire video clip $\mathcal{F}$ is input to the encoder to obtain both the current observation latent $V_1$ and the future frame latents $\{V_j\}_{j=2}^n$.

### 3.4 Unified Future Modeling

Given the current observation latent $V_1$ and the language instruction tokens $T$, our goal is to train VideoVLA to jointly predict an action chunk $\mathcal{A} = \{a_i \in \mathbb{R}^7\}_{i=1}^K$, and the future frame latents $\{V_j\}_{j=2}^n$. As illustrated in Figure 2(b), we employ a DiT-style [27] architecture, in which both the conditioning inputs and the target variables are concatenated into a unified sequence. The network is composed of a stack of self-attention blocks that model the interactions across modalities and time.

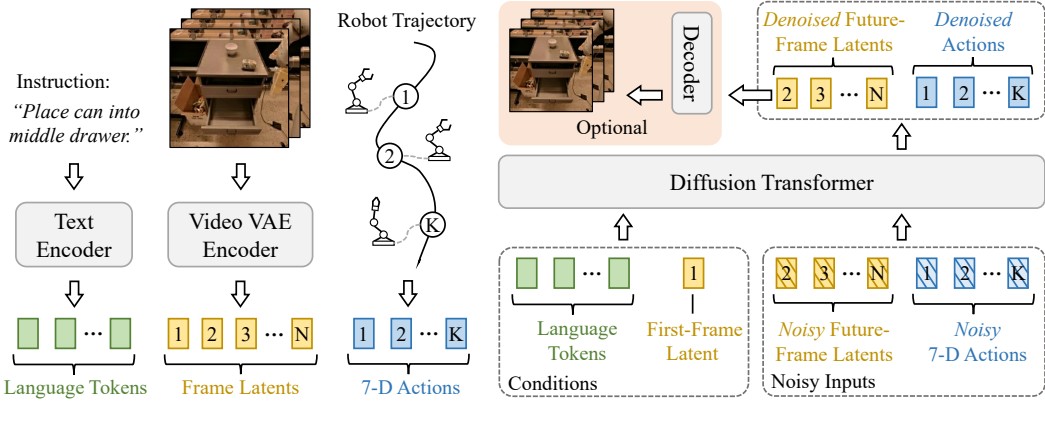

Figure 2: Overview of VideoVLA. (a) The text encoder converts the language instruction into a fixed-length token sequence, while the video encoder transforms a video clip into a sequence of frame latents, where the first latent corresponds to the first frame (i.e., the current visual observation). (b) VideoVLA adopts a Diffusion Transformer [27] architecture that conditions on the encoded language tokens and the first frame latent to jointly predict the next action chunk required to accomplish the task, along with the future frame latents that represent the anticipated visual outcomes of executing that action chunk. The video decoder, highlighted in pink, is optional and only used when visualizing the imagined future frames.

Specifically, for each visual latent—whether it represents the current observation or future frames—we flatten the spatial dimensions into a one-dimensional sequence using raster order. Let $V'_1$ denote the flattened version of the current observation latent $V_1$, and $\{V'_j\}_{j=2}^n$ denote the flattened future frame latents $\{V_j\}_{j=2}^n$. We then construct a multimodal input sequence by concatenating $T$, $V'_1$, $\{V'_j\}_{j=2}^n$, and $\mathcal{A}$. To model $\{V'_j\}_{j=2}^n$ and $\mathcal{A}$, we adopt DDPM [30]. Following its noise scheduling strategy, Gaussian noise is added to both $\{V'_j\}_{j=2}^n$ and $\mathcal{A}$. Prior to inputting into the backbone, all modalities are projected into a common embedding dimension. The model is trained to denoise the noisy versions of $\{V'_j\}_{j=2}^n$ and $\mathcal{A}$ using DDPM's diffusion loss. Following DiT [27], the noise timestep embedding is incorporated via adaptive LayerNorm. The backbone is initialized using the pre-trained CogVideoX [24] model.

## 4 Experiment

**Dataset.** We use the Open X-Embodiment (OXE) [2] dataset for pre-training, which contains over 1 million real-world robotic trajectories collected from 60 datasets spanning 22 distinct robot embodiments. Following prior works such as Octo [1], OpenVLA [3], and CogACT [4], we adopt a similar subset of OXE, comprising 22.5 million frames for pre-training. For real-world experiments, we collect a dataset consisting of 5824 samples spanning three robotic manipulation tasks: "pick", "stack", and "place". The data is collected via teleoperation using a Realman robot equipped with a 7-DoF arm and a gripper.

**Evaluation.** We conduct two types of evaluation—*in-domain* and *generalization*—across both *simulation* and *real-world* experiments. In-domain evaluation assesses scenarios where the skills (e.g., "put" and "stack") and objects (e.g., "green block") have been encountered by a specific embodiment during pre-training or fine-tuning. Generalization evaluation, on the other hand, focuses on two key capabilities: (1) executing previously learned skills on novel objects, and (2) transferring skills learned by other embodiments—yet unseen by the target embodiment—into the target embodiment.

For simulation experiments, we train our model solely on the OXE dataset, which includes data from the Google robot and WidowX robot, and evaluate on these two embodiments using the SIMPLER environment [55]. This simulation platform is designed to closely mirror real-world conditions, effectively bridging the sim-to-real gap for both control and visual inputs [55]. For real-world experiments, we further fine-tune the pre-trained model using our collected dataset.

Each task is evaluated over multiple trials in both simulation and real-world settings. Detailed setups are provided in the appendix. We report the average success rate across all trials for each experiment.

Table 1: *In-domain evaluation* of VideoVLA and prior VLA models using the WidowX robot and Google robot within the SIMPLER [55] simulation environment. All models are trained on the OXE [2] dataset. For $\pi_0$ [10], we retain only the visual observations as input, consistent with all other models in our comparison. The robot proprioception inputs are omitted because they are unavailable for most embodiments in the OXE dataset. For the WidowX robot, the four evaluated tasks are: "put spoon on towel", "put carrot on plate", "stack green block on yellow block", "put eggplant in yellow basket". We highlight the best results in **bold** and the second-best results with underline.

| Method | WidowX Robot (VM) | | | | | Google Robot (VM/VA) | | | | | Avg. (All) |
| | Put Sp. | Put Ca. | Stack Block | Put Ep. | Avg. | Pick Up Coke Can | Move Near | Open/Close Drawer | Open and Place | Avg. | |
| --- | --- | --- | --- | --- | --- | --- | --- | --- | --- | --- | --- |
| RT-1-X [2] | 0.0 | 4.2 | 0.0 | 0.0 | 1.1 | 56.7 / 49.0 | 31.7 / 32.3 | 59.7 / 29.4 | 22.2 / 11.1 | 42.7 / 30.5 | 24.8 |
| RT-2-X [2] | - | - | - | - | - | 78.7 / 82.3 | 77.9 / 79.2 | 25.0 / 35.3 | 3.7 / 20.6 | 46.3 / 54.4 | - |
| Octo-Base [1] | 15.8 | 12.5 | 0.0 | 41.7 | 17.5 | 17.0 / 0.6 | 4.2 / 3.1 | 22.7 / 1.1 | 0.0 / 0.0 | 11.0 / 1.2 | 9.9 |
| Octo-Small [1] | 41.7 | 8.2 | 0.0 | 56.7 | 26.7 | - / - | - / - | - / - | - / - | - / - | - |
| OpenVLA [3] | 4.2 | 0.0 | 0.0 | 12.5 | 4.2 | 18.0 / 60.8 | 56.3 / 67.7 | 63.0 / 28.8 | 0.0 / 0.5 | 34.3 / 39.4 | 26.0 |
| SpatialVLA [56] | 20.8 | 20.8 | 25.0 | 70.8 | 34.4 | 81.0 / 89.5 | 69.6 / 71.7 | 59.3 / **36.2** | 8.3 / 12.2 | 54.6 / 52.4 | 47.1 |
| $\pi_0$ [10] | 29.2 | **62.5** | 29.2 | **91.6** | 53.1 | 84.0 / 71.4 | 55.8 / 57.9 | 34.3 / 26.5 | 31.5 / 18.0 | 53.5 / 43.4 | 50.0 |
| CogACT [4] | 71.7 | 50.8 | 15.0 | 67.5 | 51.3 | 91.3 / 89.6 | **85.0** / **80.8** | **71.8** / 28.3 | **52.7** / 47.1 | **75.2** / 61.4 | 62.6 |
| Ours | **75.0** | 20.8 | **45.8** | 70.8 | 53.1 | **92.3** / **89.8** | 82.9 / 73.3 | 66.2 / 28.8 | 50.9 / **59.3** | 73.1 / **62.8** | **63.0** |

Table 2: *Evaluation of generalization to novel objects* using the Google robot under the SIMPLER environment. The novel objects are selected from the YCB [57] and GSO [58] datasets.

| Method | Green Cube | Carrot | Egg-plant | Wrench | Straw-berry | Plum | Tennis Ball | Cleaner Bottle | Toy Airplane | Flash-light | Avg. |
| --- | --- | --- | --- | --- | --- | --- | --- | --- | --- | --- | --- |
| OpenVLA [3] | 12.0 | 24.0 | 12.0 | 0.0 | 0.0 | 4.0 | 0.0 | 4.0 | 4.0 | 4.0 | 6.4 |
| SpatialVLA [56] | 76.0 | 52.0 | 84.0 | 32.0 | 60.0 | 56.0 | 32.0 | **56.0** | **32.0** | 28.0 | 50.8 |
| $\pi_0$ [10] | 48.0 | 60.0 | 44.0 | 16.0 | 12.0 | 44.0 | 24.0 | 16.0 | 0.0 | 24.0 | 28.8 |
| CogACT [4] | 84.0 | 72.0 | 64.0 | 16.0 | 32.0 | 44.0 | 20.0 | 32.0 | 20.0 | 40.0 | 42.4 |
| Ours | **96.0** | **84.0** | **88.0** | **40.0** | **72.0** | **80.0** | **68.0** | 44.0 | 28.0 | **52.0** | **65.2** |

**Implementation Details.** We utilize CogVideoX-5B [24] as our pre-trained backbone. By default, each inference step predicts 13 future frame latents—corresponding to 49 video frames—and 6 action steps. The model is trained for 100K iterations during pre-training and 15k iterations during fine-tuning, using 32 AMD MI300X GPUs with a batch size of 256. We employ the AdamW optimizer with a learning rate of 1e-5 and a weight decay of 1e-4. During inference, we use DDIM sampling with 50 denoising steps. For simulation experiments, we predict 13 future video latents corresponding to 49 frames, whereas for real-world experiments, we predict 4 future latents corresponding to 13 frames, for efficiency. In both settings, 6 future actions are predicted, but only the first 3 actions are executed during deployment.

### 4.1 Simulation Experiments

We use the SIMPLER [55] environment for simulation-based evaluation.

**In-Domain Evaluation.** SIMPLER offers two evaluation protocols—*Visual Matching* (VM) and *Variant Aggregation* (VA)—to assess the performance of models using the Google robot and WidowX robot. Specifically, *VM* aims to closely replicate real-world tasks by minimizing discrepancies between the simulated and real environments. *VA* builds on *VM* by introducing variations such as changes in background, lighting, and table texture. For the Google robot, both evaluation settings are available in SIMPLER, whereas for the WidowX robot, only the *VM* setting is provided. Table 1 summarizes the performance of the Google and WidowX robots under the SIMPLER environment. Our model, *VideoVLA*, achieves the highest average performance on the *VM*-WidowX-robot (averaged over 4 tasks), the highest average performance on the *VA*-Google-robot (averaged over 4 tasks), the second-highest average performance on the *VM*-Google-robot (averaged over 4 tasks), and the highest overall average performance across all 12 tasks. These results highlight VideoVLA's strong capabilities in in-domain tasks.

**Generalization to Novel Objects.** Table 2 presents the evaluation of generalization capabilities to novel objects. Specifically, we select objects from the YCB [57] and GSO [58] datasets that do not

Table 3: *Evaluation of generalization to new skills* using the Google robot within the SIMPLER environment. The new skills are transferred from the WidowX robot, as they are present in its training data but absent from the Google robot's training set. {L,R,U,B} denotes {Left, Right, Upper, Bottom}.

| Method | Put Spoon on Towel | Put Carrot on Plate | Stack Green Block on Yellow Block | Take Out of Apple | Flip Cup | Pour Coke Can | Slide to {L,R,U,B} | Avg. |
|---|---|---|---|---|---|---|---|---|
| OpenVLA [3] | 0.0 | 12.5 | 0.0 | 26.7 | 0.0 | 4.0 | 0.0 | 6.2 |
| SpatialVLA [56] | 6.3 | 31.3 | 0.0 | 31.3 | **20.0** | 40.0 | 4.0 | 18.9 |
| $\pi_0$ [10] | 18.8 | 18.8 | 0.0 | 66.7 | 8.0 | 12.0 | 4.0 | 18.3 |
| CogACT [4] | 20.8 | 41.7 | 5.0 | 43.8 | 4.0 | 20.0 | 8.0 | 20.4 |
| Ours | **56.3** | **58.3** | **20.0** | **93.8** | **20.0** | **52.0** | **40.0** | **48.6** |

Table 4: *Real-world in-domain evaluation* using the Realman robot. All models are pre-trained on the OXE dataset and subsequently fine-tuned on our collected dataset. For the "Place" task, the embodiment is required to first perform a "pick up" action followed by "placing" the object at the specified location; therefore, we report the success rates for both stages separately.

| Method | Pick Up | | | | Stack | | | Place | | | Task (All) |
|---|---|---|---|---|---|---|---|---|---|---|---|
| | Banana | Lemon | Avocado | Avg. | Cup | Bowl | Avg. | Pick Up | Place | Avg. | Avg. |
| OpenVLA [3] | 12.5 | 0.0 | 12.5 | 8.3 | 12.5 | 0.0 | 6.3 | 29.2 | 0.0 | 14.6 | 9.7 |
| SpatialVLA [56] | 37.5 | 25.0 | 50.0 | 37.5 | 25.0 | 16.7 | 20.8 | 20.8 | 0.0 | 10.4 | 22.9 |
| $\pi_0$ [10] | 62.5 | 62.5 | 75.0 | 66.7 | 58.3 | 50.0 | 54.2 | 45.8 | 16.7 | 31.3 | 50.7 |
| CogACT [4] | **75.0** | 62.5 | **87.5** | **75.0** | **83.3** | 45.8 | 64.6 | 54.2 | 16.7 | 35.5 | 58.4 |
| Ours | 62.5 | **75.0** | 75.0 | 70.8 | 75.0 | **58.3** | **66.7** | **87.5** | **25.0** | **56.3** | **64.6** |

appear in the Google robot's training data and import them into the SIMPLER environment. We evaluate the "Pick Up" skill using the Google robot on 10 novel objects: green cube, carrot, eggplant, wrench, strawberry, plum, tennis ball, cleaner bottle, toy airplane, and flashlight. Our model achieves the highest average success rate and outperforms prior models on eight out of the ten novel objects.

**Generalization to New Skills.** Table 3 presents the evaluation of skill generalization using the Google robot in the SIMPLER environment. The model is trained on the OXE dataset, which includes a diverse set of embodiments, each potentially associated with a distinct, non-overlapping set of skills. Notably, the training data for the Google and WidowX robots constitute a significant portion of the dataset, with the WidowX robot exhibiting a broader skill repertoire. To assess skill generalization, we evaluate the model's ability to transfer skills from the WidowX robot to the Google robot. Specifically, all eight skills listed in Table 3 are present in the WidowX training data but absent from the Google robot's training set. Our model outperforms the second-best model, CogACT [4], by 28.2 points in average success rate and achieves superior performance across all evaluated skills.

## 4.2 Real-World Experiments

For all experiments in this section, we fine-tune each pre-trained model—OpenVLA [3], SpatialVLA [56], CogACT [4], $\pi_0$ [10], and our VideoVLA—on our collected dataset using the Realman robot, which is equipped with a 7-DoF arm and a gripper.

**In-Domain Evaluation.** For real-world in-domain evaluation, we assess performance on three tasks: (1) Pick up the [Object] and place it onto the [Color] plate, where Object ∈ {Banana, Lemon, Avocado}, and Color ∈ {White, Blue, Yellow}; (2) Stack the [Color] [Object] into the [Color] [Object], where Object ∈ {Cup, Bowl} and Color ∈ {Pink, White, Blue, Yellow}; (3) Place the [Color] block onto the [Color] block, where Color ∈ {Red, Orange, Blue, Green, Yellow}. To increase task difficulty, we introduce novel distractor objects into the scene. As summarized in Table 4, VideoVLA achieves the highest average success rate among all evaluated models, demonstrating strong in-domain performance in real-world settings beyond simulation.

**Generalization to Novel Objects.** Table 5 presents our evaluation of real-world generalization to novel objects. Using the Realman robot, we perform the task: "Pick up the [Novel Object] and place it onto the [Color] plate", where each Novel Object is chosen from a set of 12 objects

Table 5: Evaluation of *real-world generalization to novel objects* using the Realman robot. The task is: "Pick up the [`Novel Object`] and place it onto the [`Color`] plate". B-1: an upright black bottle. B-2: a horizontally placed black bottle. B-3: an upright yellow bottle.

| Method | Blue Ball | Clear Tape | Toy Duck | Era-ser | Screw-driver | Man-go | Cab-le | Mou-se | Pea-ch | Pen | B-1 | B-2 | B-3 | Avg. |
|---|---|---|---|---|---|---|---|---|---|---|---|---|---|---|
| OpenVLA [3] | 50.0 | 0.0 | 8.3 | 25.0 | 0.0 | 0.0 | 0.0 | 0.0 | 8.3 | 8.3 | 0.0 | 8.3 | 16.7 | 9.6 |
| SpatialVLA [56] | 58.3 | 0.0 | 8.3 | 33.3 | 0.0 | 0.0 | 0.0 | 0.0 | 16.7 | 8.3 | 8.3 | 25.0 | 25.0 | 14.1 |
| $\pi_0$ [10] | 66.7 | 0.0 | 0.0 | 16.7 | 33.3 | 0.0 | 25.0 | 0.0 | 8.3 | **16.7** | **41.7** | 50.0 | 25.0 | 21.8 |
| CogACT [4] | **91.7** | 0.0 | 16.7 | **58.3** | 50.0 | 8.3 | 16.7 | 0.0 | 0.0 | 8.3 | 8.3 | 58.3 | 33.3 | 26.9 |
| Ours | 83.3 | 75.0 | 75.0 | 58.3 | 58.3 | **41.7** | **41.7** | **33.3** | **25.0** | 16.7 | 41.7 | **66.7** | **41.7** | **50.6** |

Table 6: Evaluating *real-world cross-embodiment skill transfer*: our Realman robot performs novel skills learned only by the WidowX robot, using familiar objects.

| Method | Move Block | Move Fruit | Grab Fruit | Topple Bottle | Take Out Block | Wipe Table | Avg. |
|---|---|---|---|---|---|---|---|
| OpenVLA [3] | 18.8 | 12.5 | 18.8 | 0.0 | 0.0 | 0.0 | 8.3 |
| SpatialVLA [56] | 18.8 | 25.0 | 31.3 | 6.3 | 0.0 | 0.0 | 13.5 |
| $\pi_0$ [10] | 43.8 | 31.3 | 62.5 | 12.5 | 12.5 | 8.3 | 28.5 |
| CogACT [4] | 56.3 | 50.0 | 68.8 | 18.8 | 0.0 | 16.7 | 35.1 |
| Ours | **81.3** | **68.8** | **75.0** | **43.8** | **37.5** | **41.7** | **58.0** |

Table 7: Ablation study on backbone choice. The evaluation is conducted in the SIMPLER (Visual Matching) environment using the Google robot. * denotes models trained from scratch.

| Backbone | Pick Up Coke Can | Move Near | Open/Close Drawer | Avg. |
|---|---|---|---|---|
| OpenSora-1.1 [23] | 67.7 | 57.1 | 25.9 | 50.2 |
| CogVideoX-5B* [24] | 18.6 | 10.8 | 9.2 | 12.6 |
| CogVideoX-5B [24] | **92.3** | **82.9** | **66.2** | **80.4** |

Table 8: Ablation study on the number of predicted future frames. The evaluation is conducted in the SIMPLER (Visual Matching) environment using the Google robot.

| #Frames | Pick Up Coke Can | Move Near | Open/Close Drawer | Avg. |
|---|---|---|---|---|
| 13 | 88.7 | 75.4 | 61.6 | 75.2 |
| 25 | 90.0 | 79.2 | 63.0 | 77.4 |
| 49 | **92.3** | **82.9** | **66.2** | **80.4** |

not seen during training, as listed in Table 5, and Color $\in$ {White, Blue, Yellow}. Although all evaluated models exhibit some degree of real-world generalization to novel objects, their performance varies significantly. For instance, OpenVLA [3] and SpatialVLA [56] achieve a 0% success rate on nearly half of the novel objects. In contrast, our VideoVLA successfully handles all 12 novel objects, with success rates ranging from 16.7% to 83.3%.

**Generalization to New Skills.** In this experiment, we train our model and all baseline models on a combined dataset consisting of the WidowX Robot set (derived from the OXE dataset [2]) and our own collected dataset. As a result, our model is exposed to two distinct embodiments during training: the WidowX robot and our Realman robot. These embodiments possess partially non-overlapping skill sets. To evaluate skill transfer, we focus on skills that are observed by the WidowX robot during training but never demonstrated by the Realman robot. We then assess whether the Realman robot can successfully perform these unseen skills. The set of non-overlapping skills includes: "Move", "Grab", "Topple", "Take Out", and "Wipe". The objects to be manipulated have been seen by the Realman robot during training; only the skills themselves are novel to the embodiment. As shown in Table 6, VideoVLA achieves the highest success rates across all evaluated skills. Notably, even for skills such as "Topple" and "Wipe", which differ substantially from those encountered during training, VideoVLA is still able to complete them to a certain extent.

## 4.3 Ablation Study

**Backbone.** Table 7 presents the results of two comparisons: (1) VideoVLA equipped with different pre-trained backbones, including CogVideoX-5B [24] and OpenSora-1.1 [23]; and (2) VideoVLA initialized with the pre-trained CogVideoX-5B versus VideoVLA trained from scratch. The results highlight a strong correlation between the quality of generated videos and task manipulation success rate: (1) using a higher-quality video generator, such as CogVideoX-5B, significantly improves task

Table 9: Ablation study on the dual-prediction strategy. In-domain evaluation is conducted in the SIMPLER (Visual Matching) environment using the Google robot. Generalization tests follow the settings of Tables 2 and 3.

| Method | In-domain | | | | Generalization | |
| --- | --- | --- | --- | --- | --- | --- |
| | Pick up Coke Can | Move Near | Open/Close Drawer | Avg. | Novel Objects | New Skills |
| Default | 92.3 | 82.9 | 66.2 | 80.4 | 65.2 | 48.6 |
| No video loss | 35.6 | 29.1 | 16.2 | 27.0 | 12.7 | 4.4 |
| Action only | 33.3 | 25.8 | 17.6 | 25.5 | 11.3 | 2.1 |

Table 10: Success rates for visual imagination and actual execution under generalization settings. Visual imagination outputs are evaluated by human judges to determine success or failure.

| Metric | Novel Objects | New Skills |
| --- | --- | --- |
| Visual Imagination Success Rate | 84.0 | 63.4 |
| Actual Execution Success Rate | 65.2 | 48.6 |

performance; and (2) training from scratch without pre-training on large-scale general video data results in a dramatically lower success rate.

**Time Horizon.** Table 8 presents an ablation study on how the number of predicted future frames affects robot manipulation performance. We evaluate three settings: predicting 49, 25, and 13 future frames, which correspond to 13, 7, and 4 video latents, respectively, given a video VAE encoder with a downsampling rate of 4. As the number of predicted future frames increases, the overall performance improves consistently. This suggests that having a longer temporal horizon allows the model to better anticipate the consequences of its actions, leading to more accurate execution.

**Dual-Prediction Strategy.** Table 9 presents an ablation study on our video-action-dual-prediction strategy. Three variants are compared against each other: (i) *Default*, which jointly predicts future videos and actions with denoising losses on both modalities; (ii) *No video loss*, which retains joint modeling but applies the denoising loss only to actions; and (iii) *Action only*, which predicts future actions without predicting future videos, using an action denoising loss. Both *No video loss* and *Action only* show a substantial performance decline across all tasks, underscoring the importance of the dual prediction for achieving robust task execution and strong generalization.

### 4.4 Imagination-Execution Correlation Analysis

We further analyze the correlation between the predicted visual imaginations and the actual execution outcomes. For each task, we record the video frames where the predicted actions are executed, resulting in a sequence of $M$ execution frames denoted as $\{\boldsymbol{F}_i\}_{i=1}^M$. Correspondingly, we generate $M$ related imagination frames $\{\boldsymbol{F}'_i\}_{i=1}^M$ by feeding the predicted video latents into the VAE decoder, as illustrated in Figure 2.

**Motion Similarity**. To assess the motion similarity between these two videos, we proceed as follows: For each video, we extract keypoints using SIFT [59] on the first frame, and then apply SAM [60] to segment the foreground (i.e., objects and the robot embodiment), retaining only keypoints within the foreground regions. Next, we use the SAM-PT [61] point tracking method to track these keypoints across frames. As a result, we obtain two sets of point trajectories: $\{\boldsymbol{p}_j\}_{j=1}^P$ from the actual execution and $\{\boldsymbol{p}'_j\}_{j=1}^{P'}$ from the visual imagination, where each $\boldsymbol{p}_j$ and $\boldsymbol{p}'_j$ represents a trajectory of a foreground keypoint. We then apply the Hungarian matching algorithm to identify correspondences between trajectories in the two sets. For each matched pair, we compute the normalized cosine similarity between the trajectory vectors. Finally, we report the *mean* of all pairwise similarities as the overall metric for robot motion similarity between the predicted visual imaginations and the actual executions. The results in Figure 3 show that a higher robot motion similarity between the visual imaginations and the actual executions corresponds to a higher task success rate. This indicates that (1) there is a strong correlation between the predicted visual imaginations and actual executions, and (2) predicting accurate future visual outcomes facilitates successful robotic manipulation.

**Task Performance Comparison**. We also compare the quality of visual imagination and actual execution results in the generalization settings in Table 10. Since visual imaginations are video

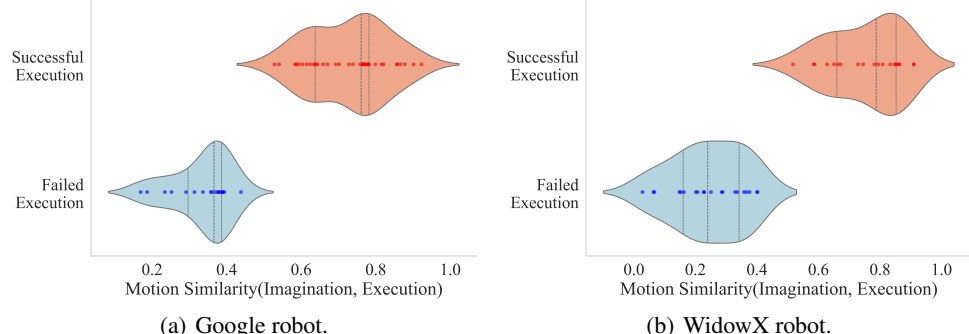

(a) Google robot.             (b) WidowX robot.

Figure 3: Each sub-figure illustrates the relationship between robot motion similarity—comparing visual imaginations with actual executions—and task success. Each point represents either a successful or failed execution. A higher robot motion similarity corresponds to an increased likelihood of successful execution. The plots show aggregated statistics across tasks in the SIMPLER environment using (a) the Google robot and (b) the WidowX robot.

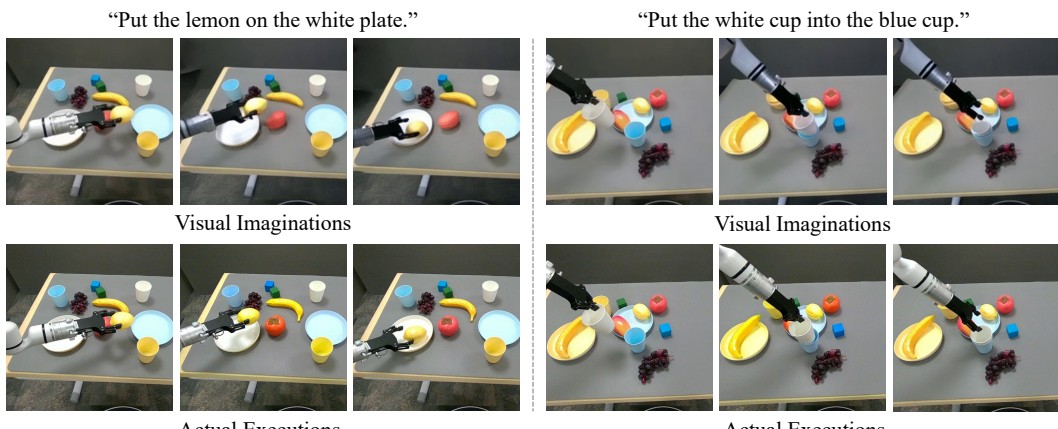

Figure 4: Visualization of VideoVLA's predicted visual imaginations and corresponding real-world executions during task completion, demonstrating a strong correlation between imagined and actual outcomes. Additional visualizations are provided in the appendix.

outputs which cannot be automatically evaluated for success, we conduct human evaluation and deem an imagined trajectory successful if it (i) follows the instruction semantically and (ii) exhibits no salient geometric distortions or violations of physical plausibility. We observe that most visual imaginations yield reasonable outcomes, attaining 84.0% success on novel objects and 63.4% on new skills. As expected, actual execution achieves lower success (65.2% and 48.6%, respectively), reflecting the added difficulty of precise physical grounding, actuation noise, and perception errors in real environments. Additionally, we provide two qualitative visualizations in Figure 4.

## 5   Conclusion

In this paper, we present *VideoVLA*, a vision-language-action framework that leverages large pre-trained video generators—unlike prior approaches that rely primarily on large perception or vision-language models. VideoVLA jointly predicts future actions to be executed and generates visual imaginations depicting the anticipated outcomes of those actions. Experimental results reveal a strong correlation between imagined futures and actual execution: high-quality visual predictions are consistently associated with reliable action plans and task success. VideoVLA not only demonstrates strong performance on tasks involving seen objects and previously learned skills, but also exhibits remarkable generalization to novel objects and cross-embodiment skill transfer in both simulated and real-world environments. These capabilities stem from the power of large-scale pre-trained video generation and our dual-prediction strategy. Our findings highlight the potential of generative video models as a scalable foundation for general-purpose robot manipulation.

## Acknowledgments

We thank the reviewers for their valuable comments. This work was partly supported by National Natural Science Foundation of China under grant No. 62088102.

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

# Appendix

## A  Evaluation Details

We report the number of trials for each experiment in both simulation and real-world evaluations. For each model, including our VideoVLA and the baselines, we strictly follow a consistent evaluation protocol.

**Simulation Experiments.** Table 11 lists the number of trials for each experiment conducted in the SIMPLER environment [55], including Google robot experiments in SIMPLER Visual Matching (VM) and Variant Aggregation (VA), WidowX robot experiments in SIMPLER Visual Matching (VM), and generalization experiments with the Google robot.

**Real-World Experiments.** Table 12 presents the number of trials conducted for each real-world experiment.

Table 11: The number of trials for each simulation experiment.

| Setting | Task | # Trials |
|---|---|---|
| Google Robot (SIMPLER-VM) | Pick Up Coke Can | 300 |
| | Move Near | 240 |
| | Open/Close Drawer | 216 |
| | Open Top Drawer and Place Apple | 108 |
| Google Robot (SIMPLER-VA) | Pick Up Coke Can | 825 |
| | Move Near | 600 |
| | Open/Close Drawer | 378 |
| | Open Top Drawer and Place Apple | 189 |
| WidowX Robot (SIMPLER-VM) | Put Spoon on Towel | 24 |
| | Put Carrot on Plate | 24 |
| | Stack Green Block on Yellow Block | 24 |
| | Put Eggplant in Yellow Baske | 24 |
| Google Robot (SIMPLER) | Each Novel Object | 25 |
| Google Robot (SIMPLER) | Each New Skill | 20 |

Table 12: The number of trials for each real-world experiment.

| Task | # Trials |
|---|---|
| Pick Up | 24 |
| Stack | 48 |
| Place | 24 |
| Each Novel Object | 12 |
| Each New Skill | 16 |

## B  More Analysis

**Causal masking vs. bidirectional attention.** To further examine whether a causal masking strategy offers advantages over our default bidirectional approach, we conduct an ablation in which a causal mask is applied: action tokens can attend to video tokens, but video tokens cannot attend to action tokens. As reported in Table 13, we observe a consistent drop in success rate with causal masking relative to the default bidirectional model. These results indicate that allowing video tokens to access action tokens helps the model better capture how specific actions drive physical changes in the visual scene. Furthermore, bidirectional interaction between action and vision enhances coherence and alignment—allowing actions to more effectively guide video prediction, while video prediction, in turn, supports more accurate action prediction.

Table 13: Ablation study on causal masking strategy. The evaluation is conducted in the SIMPLER (Visual Matching) environment using the Google robot.

| Method | Pick up Coke Can | Move Near | Open/Close Drawer | Average |
|---|---|---|---|---|
| Default | 92.3 | 82.9 | 66.2 | 80.4 |
| Causal mask | 89.3 | 76.2 | 61.1 | 75.5 |

Table 14: Ablation study on diffusion schedules. The evaluation is conducted in the SIMPLER (Visual Matching) environment using the Google robot.

| Diffusion Schedule | Pick Up Coke Can | Move Near | Open/Close Drawer | Avg. |
|---|---|---|---|---|
| Sync-training, Sync-inference (Default) | 92.3 | 82.9 | 66.2 | 80.4 |
| Async-training, Sync-inference | 87.3 | 74.1 | 60.2 | 73.8 |
| Async-training, Async-inference | 84.7 | 70.8 | 57.4 | 71.0 |

**Asynchronous noising and inference.** By default, our model applies a *synchronous diffusion schedule* to both vision and action tokens during training and inference, which means that these two modalities share the same diffusion timestep and strategy. To further examine the impact of alternative schedules on model behavior, we evaluate two variants that decouple the action and video noising schedules during training: (i) *asynchronous training, synchronous inference*, which uses different noising schedules in training but jointly denoises actions and video at test time; and (ii) *asynchronous training, asynchronous inference*, which at test time first denoises video latents (stage-1) and then conditions on the denoised video to generate actions (stage-2). As demonstrated in Table 14, both variants exhibit inferior performance compared to the default joint (synchronous) strategy across all tasks, with average scores of 73.8% and 71.0% versus 80.4%, reflecting drops of 6.6 and 9.4 percentage points, respectively. We hypothesize that the degradation arises because actions and video are temporally aligned modalities; joint training and denoising allow cross-modal information to be exchanged at every step, yielding complementary supervision.

## C   More Visualizations

Figures 5 and 6 show visualizations of VideoVLA's predicted visual imaginations alongside the corresponding executions during task completion, for real-world and simulation experiments, respectively.

## D   Limitations and Broader Impacts

One key limitation of this work is its inference speed. For real-world deployment, VideoVLA predicts 4 future latents (corresponding to 13 video frames) and 6 future actions (of which the first 3 are executed), using DDIM denoising with 10 denoising steps. This process takes approximately 1.1 seconds on a single H100 GPU, resulting in an effective control frequency of around 3 Hz. This limitation primarily stems from the reliance on a large pre-trained video generator—CogVideoX-5B in our case. Nonetheless, our work demonstrates the feasibility of leveraging large-scale pre-trained generation models, rather than the commonly used pre-trained perception and understanding models, to enable end-to-end vision-language-action (VLA) robotic manipulation. Future directions for improving VideoVLA's inference speed include pre-training a robot-oriented video generator with a smaller model size than general-purpose generators, adopting one-step denoising techniques such as the ShortCut model [62], and applying model distillation. We leave inference acceleration as an important avenue for future exploration.

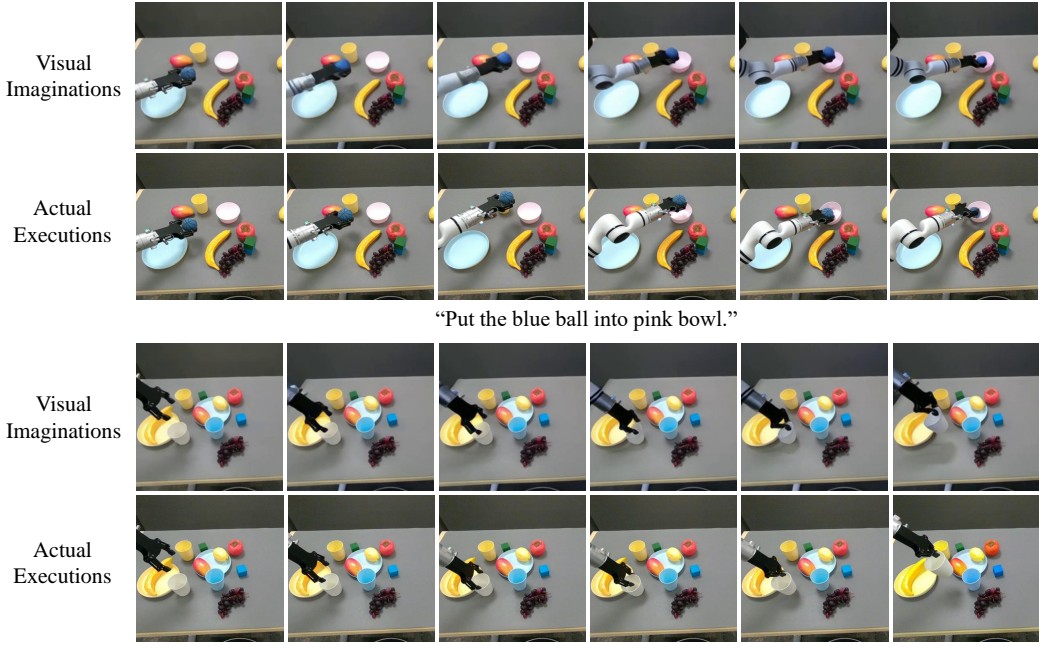

Figure 5: Visualizations of VideoVLA's predicted visual imaginations and the corresponding executions during task completion in real-world experiments.

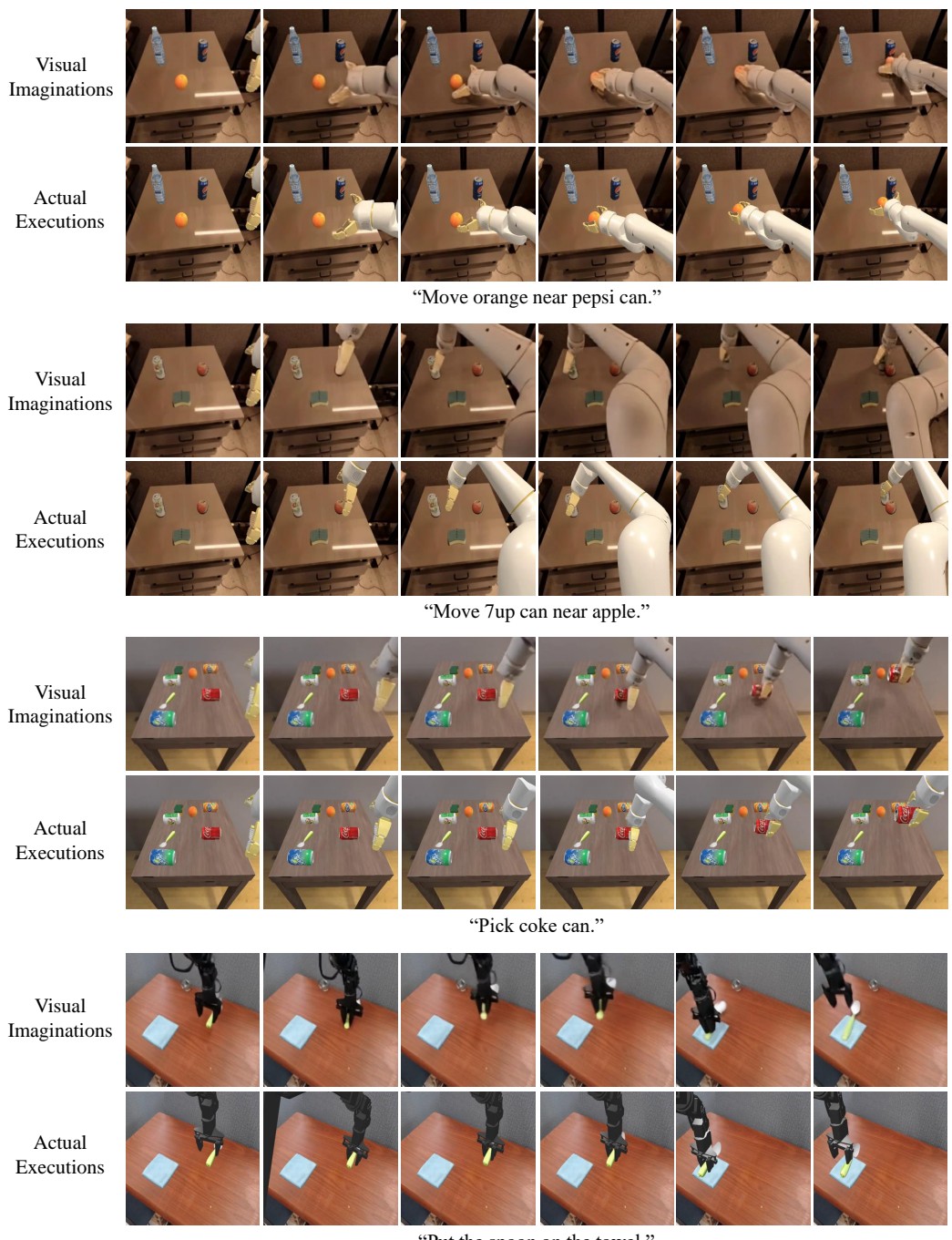

Figure 6: Visualizations of VideoVLA's predicted visual imaginations and the corresponding executions during task completion in simulation experiments.

