# OpenReview forum: "VideoVLA: Video Generators Can Be Generalizable Robot Manipulators"
_NeurIPS.cc/2025/Conference — NeurIPS 2025 poster_

### Official Review · Reviewer_cgvg · 2025-07-01

**Clarity:** 2
**Significance:** 2
**Originality:** 1
**Rating:** 4
**Confidence:** 4

**Summary:**

This paper proposes Roboseer which leverages diffusion models to output actions and future outcomes at the same time.

**Questions:**

see above

**Ethical Concerns:**

["NO or VERY MINOR ethics concerns only"]

**Final Justification:**

I wanna emphasize again that it's not that I did not appreciate the merits of the paper, but that the paper over-claimed too much and missed potentially paragraphs of related works - this is a serious major problem of this paper. this also makes it hard to grasp the true contributions of this paper.

However, the authors discussed these related works and their differences in the rebuttal phase. I assume that they will add all these to their final version and revise their paper so it does not over-claim; and therefore will raise the point to 4.

**Limitations:**

see above

**Quality:**

2

**Strengths And Weaknesses:**

* There are a lot of experiments

Weaknesses:
I seriously doubt that the authors had zero knowledge of using video generation models for robot manipulation, for example, those works that use inverse dynamics. They did not mention those works in the related works section, and not compare to them in the experiments section as well. They even claimed:

*this is the first work that adapts a large-scale video generator to the domain of robotic manipulation*

Such papers, not to say hundreds, but at least tens of papers are in this field (but really I think there are already hundreds of such papers). To name a few:

Learning Interactive Real-World Simulators
Learning to Act from Actionless Videos through Dense Correspondences
Learning Universal Policies via Text-Guided Video Generation
Video Language Planning
GEVRM: Goal-Expressive Video Generation Model For Robust Visual Manipulation
Unleashing Large-Scale Video Generative Pre-training for Visual Robot Manipulation
TASTE-Rob: Advancing Video Generation of Task-Oriented Hand-Object Interaction for Generalizable Robotic Manipulation
Gen2Act: Human Video Generation in Novel Scenarios enables Generalizable Robot Manipulation
Unleashing Large-Scale Video Generative Pre-training for Visual Robot Manipulation
GR-2: A Generative Video-Language-Action Model with Web-Scale Knowledge for Robot Manipulation
RoboDreamer: Learning Compositional World Models for Robot Imagination
VidMan: Exploiting Implicit Dynamics from Video Diffusion Model for Effective Robot Manipulation

Of course the authors can say that there is at least some level of novelty, even not prominent novelty, which distinguishes themselves from these papers. But the authors did not mention any of these papers and discuss about the differences so we are unable to tell as well.

---

> ### Author Rebuttal · Authors · 2025-07-31
>
> Dear Reviewer cgvg,
>
> We appreciate your feedback to our work. Please find our responses to your questions below.
>
> **Q1: I seriously doubt that the authors had zero knowledge of using video generation models for robot manipulation, for example, those works that use inverse dynamics. They did not mention those works in the related works section, and not compare to them in the experiments section as well. They even claimed: this is the first work that adapts a large-scale video generator to the domain of robotic manipulation.**
>
> Thank you for highlighting these previous works. In our paper, we propose a new large vision-language-action (VLA) model and primarily compare it with existing end-to-end VLA models such as RT-2, Pi0, CogACT, SpatialVLA, as shown in Table 1 of the main paper.
>
> Below, we first provide a discussion of the papers you mentioned, which will also be incorporated into the revised version of our paper. We then outline the key differences between our work and these prior approaches, followed by the rationale for not including direct comparisons. Finally, we describe our revision plan to better integrate these related works.
>
> **Discussion of the Mentioned Papers.**
>
> Several works such as [1,2,3] train video generation models on robot video data and extract actions using an inverse dynamics model based on adjacent frame pairs. However, these inverse dynamics models are typically trained with limited data and simple architectures, making them less robust to changes in environment appearance (e.g., lighting, background, textures). Moreover, extracting actions from only two frames often fails to capture fine-grained or complex physical dynamics.
>
> The work of [4] estimates end-effector actions by computing optical flow between frames and leveraging depth maps. However, since the gripper’s motion often produces weak optical flow signals, the authors resort to heuristic rules (e.g., large z-axis motion) to infer gripper closure, which can lead to inaccuracies in predicting precise gripper actions.
>
> Other approaches [5,6] extract a goal frame from the predicted video and use it as input to a goal-image-conditioned policy for action prediction, which introduces additional training and engineering overhead due to the need for a separately trained goal-image-conditioned policy.
>
> Gen2Act [7] and TASTE-Rob [8] focus on human hand–object interaction video generation, where actions are produced via a human-to-robot transfer module based on synthesized human hand trajectories.
>
> GR-1 [9] and GR-2 [10] generate video frames autoregressively, predicting one frame at a time. This design limits their capacity for global planning and foresight. Moreover, they do not leverage pre-trained video foundation models, thereby missing the benefits of large-scale visual pretraining for generalization.
>
> VidMan [11] inserts implicit inverse dynamics adapters into a video diffusion transformer to integrate future frame information and then applies a separate diffusion-based action head for action generation. In their framework, the video and action modalities are trained in two stages without joint training.
>
> **Key Differences from the Mentioned Works and Rationale for Not Including Direct Comparisons.**
>
> ***The key difference of our approach compared to prior works lies in the design of a unified, end-to-end vision-language-action (VLA) model directly adapted from a large pre-trained video generation model. Unlike modular approaches, our method jointly predicts the video and action modalities within a single large transformer architecture, and it leverages the generalization capabilities learned from video generation pretraining on massive real-world videos. No such model exists to our knowledge.***
>
> In contrast, most of the mentioned works rely on additional modules to produce final actions. As mentioned above, [1–3] introduce separate inverse dynamics modules. [4] employs hand-crafted heuristic rules to generate action trajectories. [5–6] require an additional module to infer final actions from goal images generated in a separate stage. [7,8] first train models on human-object interaction data and then design task-specific algorithms to transfer the learned human-centric representations to robot action prediction. [11] gathers future frame information by querying features from video generation model and then uses a separate diffusion action head to generate action. The work of [9] and [10] do not generate video chunks as global planning and foresight; nor do they leverage pre-trained video foundation models.
>
> In our method comparison, we focus exclusively on end-to-end VLA models and ensure fair comparison by reproducing open-sourced baselines such as OpenVLA, Pi0, CogACT, SpatialVLA. We consider these recent large-scale VLA models to be the primary competitors to our approach. Among the works mentioned by the reviewer, [9-11] are the more related as they fit better to the category of end-to-end VLA models. However, none of these works have released training code, making it infeasible to compare with these methods in our experimental setup.
>
> **Plans to Incorporate the Mentioned Papers in the Revised Version.**
>
> Thank you again for pointing out these excellent works. We will incorporate a detailed discussion of the mentioned related studies in the revised version of the paper to provide a more comprehensive comparison and context. Specifically, we plan to add a paragraph in the **Related Work** section to discuss prior efforts that apply video prediction to enhance action policies. A preliminary draft is provided below and we will expand it to include additional relevant works:
>
> > **Video Generation for Robot Manipulation.** The use of video generation models for robot manipulation has been an active area of research. Most existing approaches incorporate video generation as a visual planning component within modular frameworks to assist with action prediction. For instance, [1–3] extract actions from video predictions using inverse dynamics models. [4] estimates end-effector actions by computing optical flow between frames and leveraging depth maps. Approaches like [5,6] extract a goal frame from the predicted video and use it as the condition for action prediction. [7] and [8] focus on human hand–object interaction video generation and design models for human-to-robot transfer. [11] gathers future frame information by querying features from video generation model and uses a diffusion head to generate action. [9] and [10] generate video frames autoregressively and predict one frame at a time along with an action. Different from these works, our method introduces a unified, end-to-end VLA model directly adapted from a large pre-trained video generation model. Unlike modular approaches, our method jointly predicts the video and action modalities within a single large transformer architecture, and it leverages the generalization capabilities learned from video generation pretraining on massive real-world videos.
>
> Please let us know if this would be an acceptable solution.
>
> **Reference**
>
> [1]  Learning Universal Policies via Text-Guided Video Generation Video Language Planning
>
> [2]  LEARNING INTERACTIVE REAL-WORLD SIMULATORS
>
> [3]  RoboDreamer: Learning Compositional World Models for Robot Imagination
>
> [4]  Learning to Act from Actionless Videos through Dense Correspondences
>
> [5] VIDEO LANGUAGE PLANNING
>
> [6] GEVRM: Goal-Expressive Video Generation Model For Robust Visual Manipulation
>
> [7] Gen2Act: Human Video Generation in Novel Scenarios enables Generalizable Robot Manipulation
>
> [8] TASTE-Rob: Advancing Video Generation of Task-Oriented Hand-Object Interaction for Generalizable Robotic Manipulation
>
> [9] Unleashing Large-Scale Video Generative Pre-training for Visual Robot Manipulation
>
> [10] GR-2: A Generative Video-Language-Action Model with Web-Scale Knowledge for Robot Manipulation
>
> [11] VidMan: Exploiting Implicit Dynamics from Video Diffusion Model for Effective Robot Manipulation
>
> ---
>
> Once again, thank you for your valuable time and effort in reviewing our work. Please let us know if you have further questions.
>
> Authors of Paper 7746

---

> > ### Comment · Reviewer_cgvg · 2025-08-04
> >
> > I wanna emphasize again that it's not that I did not appreciate the merits of the paper, but that the paper over-claimed too much and missed potentially paragraphs of related works - this is a serious major problem of this paper. this also makes it hard to grasp the true contributions of this paper.
> >
> > However, the authors discussed these related works and their differences in the rebuttal phase. I assume that they will add all these to their final version and revise their paper so it does not over-claim; and therefore will raise the point to 4.

---

> > > ### Author Response · Authors · 2025-08-05
> > >
> > > Thank you for your suggestions and additional comments! We're glad to see that our clarification addresses your concern. Yes, we commit that a paragraph will be added into the related work section to discuss the aforementioned related works, highlighting our differences to them and providing a more precise articulation of our contributions. Thanks again.
> > >
> > > Authors

---

### Official Review · Reviewer_Knht · 2025-07-02

**Clarity:** 3
**Significance:** 3
**Originality:** 2
**Rating:** 4
**Confidence:** 4

**Summary:**

The authors propose a method of incorporating action diffusion to a pretrained Video Diffusion model (CogvideoX) to jointly train on action and video generation for robotics tasks. The authors demonstrate significant generalization compared to existing methods like OpenVLA and $\pi_0$ in both simulation and real-world experiments.

**Questions:**

see weaknesses

**Ethical Concerns:**

["NO or VERY MINOR ethics concerns only"]

**Final Justification:**

The authors demonstrate significant generalization compared to existing methods and have also ablated different variations of their architecture and formulation showing consistently good results.

**Limitations:**

yes

**Quality:**

2

**Strengths And Weaknesses:**

Strengths

1. The authors propose a new method that can leverage pretrained video diffusion models like CogVideoX to additionally generate actions to solve robotics tasks.
2. The authors demonstrate huge improvements in generalization to new objects and skills in both simulation and real-world tasks.


Weaknesses :

1. It is not clear how much of the generalisation stems only from the pre-trained knowledge of the CogVideo model and how much from the joint modelling proposed by the authors. Two important ablations here are training from scratch without the cog weights and also a variant that only predicts (and is trained on) action instead of both video and action, using the pre-trained cogvideo weights.

2. A very important generalization that should also be evaluated is generalisation to unseen environments, since this is a very hard challenge in robotics. It would be interesting to see how the current method performs in this setup.

3. There needs to be ablations using a different noising schedule for the actions and video latents. One advantage of such a mechanism is that this would allow conditional prediction of both video based on action and action based on video by setting the corresponding denoising step to the min noise for action's or video's schedule while denoising only video or action respectively.

4. One very interesting effect of being able to predict both video and actions, like in this case, allows us to visualize failure cases to see what the model "expects to happen in the scene" using such actions. Some analysis of failure cases inspecting the generated videos should be added to see if there are patterns to be noticed/insights to be gained from them about the cause of the failure.

4. In L196-197the authors mention they predict 16 latents corresponding to 49 frames. This is not possible with a downsampling factor of 4, but 13 (=(49-1)/4 + 1) latents as mentioned elsewhere in the paper. If this is not a typing mistake, then the authors need to explain this in detail.

---

> ### Author Rebuttal · Authors · 2025-07-31
>
> Dear Reviewer Knht,
>
> We sincerely appreciate your thoughtful feedback. Please find our responses to your questions below.
>
>
> **Q1: It is not clear how much of the generalisation stems only from the pre-trained knowledge of the CogVideo model and how much from the joint modelling proposed by the authors. Two important ablations here are training from scratch without the cog weights and also a variant that only predicts (and is trained on) action instead of both video and action, using the pre-trained cogvideo weights.**
>
>
>
> **Q1.1: Effect of using pre-trained CogVideo weights.**
>
> As shown in Table 7 and Lines 268–269, training the model from scratch without using the pre-trained CogVideo weights leads to a substantial performance drop—from 80.4\% to 12.6\%. This clearly highlights the importance of the prior knowledge embedded in CogVideo, which serves as a strong foundation for generalization and is critical for achieving high downstream performance.
>
> **Q1.2: Effect of joint video-action modeling.**
>
> To further demonstrate the effectiveness of the proposed dual-prediction strategy, we compare three variants of our method:
>
> 1. *Default* – jointly predicts future videos and actions, with denoising losses applied to both modalities;
>
> 2. *No video loss* – jointly models videos and actions but applies the denoising loss only to actions;
>
> 3. *Action only* – predicts future actions only, without predicting future videos, and applies a denoising loss on actions.
>
> We evaluate these three variants under two experimental settings:
>
> 1. In-domain evaluation in the SIMPLER VM environment: This setting assesses in-domain performance. The experimental configuration follows that of Table 1 in the main paper. Results are shown in the table below:
>
> | Method | Pick up Coke Can | Move Near |Open/Close Drawer | Average |
> |:------|:--------:|:-------:|:-------:|:-------:|
> | *Default* |   92.3      |     82.9  |     66.2  | 80.4 |
> | *No video loss* |   35.6      |     29.1  |     16.2 | 27.0 |
> | *Action only* |   33.3      |     25.8  |     17.6 | 25.5
>
> 2. Generalization evaluation: This setting examines performance on novel objects and new skills, using the same configuration as Tables 2 and 3 in the main paper. Results are presented below:
>
> | Method | Novel Objects | New Skills |
> |:------|:--------:|:-------:|
> | *Default* |   65.2      |     48.6  |
> | *No video loss* |   12.7      |    4.39  |
> | *Action only* |   11.3      |    2.19  |
>
>
> Both ablated variants— *No video loss* and *Action only*—show a substantial performance decline across all tasks. These results underscore the importance of the full dual-prediction framework, which is essential for achieving strong generalization and robust task execution.
>
> We will include these studies in the revised paper.
>
> **Q2: A very important generalization that should also be evaluated is generalisation to unseen environments, since this is a very hard challenge in robotics. It would be interesting to see how the current method performs in this setup.**
>
> As noted in Lines 205–206 of the main paper, the SIMPLER (VA) benchmark [1] introduces substantial domain variations, including changes in background, camera viewpoint, lighting conditions, and table textures—collectively referred to as "environment variants" [1]. These variations differ significantly from the original training distribution. As shown in Table 1 of the main paper, our model achieves the best performance under the SIMPLER (VA) setting, demonstrating its robustness and strong generalization ability to novel environments.
>
> Furthermore, as illustrated in the third row of Figure 2 in the supplementary material, the SIMPLER (VA) setting includes previously unseen tables and backgrounds not present in the original Google Robot dataset. Despite these domain shifts, our model maintains high-quality visual imagination and preserves consistency between imagined and actual executions. These results indicate that RoboSeer is capable of effective domain transfer across diverse and challenging visual environments.
>
> **Q3: There needs to be ablations using a different noising schedule for the actions and video latents. One advantage of such a mechanism is that this would allow conditional prediction of both video based on action and action based on video by setting the corresponding denoising step to the min noise for action's or video's schedule while denoising only video or action respectively.**
>
> Following your suggestions, we tried two variants of our model with different noising schedules for action and video latent.
>
> 1. *Asynchronous training, synchronous inference*: This variant adopts different noising schedules for actions and video latents during training. However, during inference, both are denoised synchronously, as in our default method.
>
> 2. *Asynchronous training, asynchronous inference*: This variant also uses different noising schedules during training. At inference time, we first input the noisy actions and video latents but denoise only the video latents (stage-1). The resulting denoised video latents are then combined with the original noisy actions to denoise the actions (stage-2).
>
> We evaluate three models under the SIMPLER VM environment. The experimental configuration follows that of Table 1 in the main paper. Results are shown in the table below:
>
> | Method | Pick up Coke Can | Move Near |Open/Close Drawer | Average |
> |:------|:--------:|:-------:|:-------:|:-------:|
> | *Default*  |   92.3      |     82.9  |     66.2  | 80.4 |
> | *Asynchronous training, synchronous inference*|   87.3    |     74.1  |    60.2 |  73.8 |
> | *Asynchronous training, asynchronous inference*|   84.7    |     70.8  |    57.4 | 71.0 |
>
> We observe that asynchronous training leads to a performance drop across all tasks compared to our default method. This is likely because actions and video are temporally aligned, synchronous modalities. Joint training and denoising allow the two modalities to complement each other, similar to video-audio generation models where both modalities are typically modeled jointly to enhance performance.
>
> Under the *asynchronous inference* setting, we observe that the denoised video latents from stage-1 are often of decent quality and semantically consistent with real-world expectations. When using these denoised video latents as conditions to generate actions in stage-2, the resulting actions still yield reasonably high performance (i.e., the third row in the table above). This indicates that the model trained with asynchronous noising schedules indeed possesses a decent degree of video-conditioned action generation capability.
>
> We are unsure if this satisfactorily addresses your questions, particularly regarding the "advantage" of allowing conditional prediction of both video based on action and action based on video. Please let us know if it doesn't or you have any further questions.
>
> **Q4: Some analysis of failure cases inspecting the generated videos should be added to see if there are patterns to be noticed/insights to be gained from them about the cause of the failure.**
>
> We thank the reviewer for this insightful suggestion. We conducted an analysis of failure cases in which the visual imaginations were either inaccurate or visually unpleasant, and found that such failures often lead to incorrect action predictions. Specifically, we randomly analyzed 60 failure cases where the generated videos failed to reflect the instruction intent, exhibited physically implausible transitions, or appeared visually incoherent. Among these, 55 episodes (91.7\%) resulted in failed executions. This indicates that inaccurate visual imagination is a major contributor to incorrect action prediction.
>
> We indeed observed interesting pattens from the visual imagination which can explain the execution failures. For example, when the gripper is not visible in the initial frame, the visual imagination may plan a motion trajectory not consistent with the robot's physical state, leading to execution failure. This issue can be handled by incorporating robot state into the model in our future work. In a few cases where the robot confuses the target objects to be grasped with other similarly colored ones, the visual imagination is found to have the same confusion in the generated videos.
>
> Due to rebuttal constraints, we are unable to provide examples here. However, in the revised version of the paper, we will include additional examples of both visual imagination and actual execution in failure cases.
>
> **Q5: In L196-197 the authors mention they predict 16 latents corresponding to 49 frames. This is not possible with a downsampling factor of 4, but 13 (=(49-1)/4 + 1) latents as mentioned elsewhere in the paper. If this is not a typing mistake, then the authors need to explain this in detail.**
>
> We thank the reviewer for carefully pointing this out. This was indeed a typographical error—the model predicts 13 vision latents corresponding to 49 frames. We will correct this in the revised version of the paper. Other instances in the paper referencing 13 latents corresponding to 49 frames, such as Line 192, are correct.
>
>
> **Reference**
>
> [1] Evaluating Real-World Robot Manipulation Policies in Simulation (SIMPLER)
>
> *Once again, thank you for your valuable time and effort in reviewing our work.*
>
> Authors of Paper 7746

---

> > ### Comment · Reviewer_Knht · 2025-08-05
> >
> > The authors have addressed my concerns and therefore I will maintain my positive rating. It would be nice to see the above results in the final version of the paper

---

> > > ### Author Response · Authors · 2025-08-06
> > >
> > > Thank you for the comment. We're glad to see that your concerns are addressed and will incorporate the above results into the revised paper. Thanks again for insightful suggestions.
> > >
> > > Authors

---

### Official Review · Reviewer_4hpC · 2025-07-03

**Clarity:** 4
**Significance:** 3
**Originality:** 3
**Rating:** 5
**Confidence:** 5

**Summary:**

This paper proposes RoboSeer, a novel vision-language-action (VLA) framework that leverages pre-trained video generation models—rather than understanding models—for generalizable robot manipulation. RoboSeer adopts a dual-prediction strategy, simultaneously forecasting future actions and their visual consequences using a multi-modal diffusion transformer. By conditioning on current visual observations and language commands, RoboSeer predicts the next action chunk and generates imagined future frames. The model is trained using a DDPM-style diffusion loss and initialized with a large-scale video generator (CogVideoX-5B). Experiments in both simulation and real-world environments demonstrate state-of-the-art performance on in-domain tasks, strong generalization to unseen objects, and cross-embodiment skill transfer.

**Questions:**

- Efficiency: Can RoboSeer skip visual imagination inference when only actions are needed (e.g., for low-latency execution)?
- Failure Modes: What happens when visual imagination is incorrect? Is action prediction always degraded?
- Domain Transfer: Have you tried generalizing to new environments (e.g., different lighting or backgrounds)?
- Release Plans: Will the model, code, and pre-trained weights be made available?
- Smaller Models: How does RoboSeer perform with smaller video generators (e.g., CogVideoX-B)?

**Ethical Concerns:**

["NO or VERY MINOR ethics concerns only"]

**Final Justification:**

This paper is novel in robotics manipulation, meets SOTA, and is solid in experiment. Therefore, I confirm my rating as 5.

**Limitations:**

yes

**Paper Formatting Concerns:**

good

**Quality:**

3

**Strengths And Weaknesses:**

# Strengths
- Paradigm Shift: Moves from "understanding" to "generation" in robot policy learning.
- Dual-Prediction Strategy: Joint action and visual outcome prediction improve accuracy and interpretability.
- Comprehensive Experiments: Evaluation across multiple embodiments, domains, and tasks, including real-world deployment.
- Strong Generalization: Outperforms baselines like π0, CogACT, and OpenVLA on novel objects and unseen skills.
- Correlation Analysis: Demonstrates that visual imagination quality correlates with actual task success.
- Ablation Studies: Carefully explores ithe mpact of video generator quality and prediction horizon.
# Weaknesses
- Lacks Theory: No theoretical analysis explaining why dual-prediction improves generalization.
- High Compute Cost: Requires 32×192GB GPUs and large-scale training.
- Visual Imagination Dependency: Unclear how much visual prediction helps over action-only baselines.
- Skill Transfer Scope: Skill generalization tested on a limited set of tasks; broader testing would strengthen results.

---

> ### Author Rebuttal · Authors · 2025-07-29
>
> Dear Reviewer 4hpC,
>
> We sincerely appreciate your valuable feedback. Please find our responses to your questions below.
>
>
> **Q1: Lacks Theory: No theoretical analysis explaining why dual-prediction improves generalization.**
>
> Thank you for raising this question. We now provide the insight pertaining to why dual prediction improves generalization.
>
> First, our training data consists of temporally aligned robot manipulation videos and their corresponding action sequences. Our model learns the cross-modal correspondence—particularly temporal alignment—between the visual and action modalities, enabling it to jointly predict both. Each modality provides complementary information to the other.
>
> Second, we leverage large-scale pre-trained video generators with strong generalization capabilities in video generation—i.e., the ability to generate novel content during inference. Our model can be viewed as a form of post-training on top of these video generators, allowing it to inherit their generalization capacity. As a result, the model can imagine plausible visual executions for novel environments, skills, and objects. Since it has already learned the alignment between imagined visuals and corresponding actions, this generalization ability transfers to the action domain, leading to improved generalization in robot manipulation tasks.
>
>
> To further demonstrate the effectiveness of the proposed dual-prediction strategy, we compare three variants of our model:
>
> 1. *Default* – jointly predicts future videos and actions, with denoising losses applied to both modalities;
>
> 2. *No video loss* – jointly models videos and actions but applies the denoising loss only to actions;
>
> 3. *Action only* – predicts future actions only, without predicting future videos, and applies a denoising loss on actions.
>
> We evaluate the performance of these models in the SIMPLER environment, focusing on generalization to novel objects and new skills. The experimental setup follows the same configuration as Tables 2 and 3 in the main paper. The results are presented in the table below:
>
>
> | Method | Novel Objects | New Skills |
> |:------|:--------:|:-------:|
> | *Default*  |   65.2      |     48.6  |
> | *No video loss* |   12.7      |    4.39  |
> | *Action only* |   11.3      |    2.19  |
>
> Both ablated variants— *No video loss* and *Action only* —exhibit a significant drop in generalization performance, especially on tasks involving novel objects and new skills. These results highlight the importance of the full dual-prediction framework, which plays a critical role in enabling robust generalization.
>
> We will include this analysis in the revised paper and thank you for the thoughtful suggestion.
>
>
> **Q2: High Compute Cost: Requires 32×192GB GPUs and large-scale training.**
>
> VLA models typically require large-scale training data and sizable network architectures to achieve robust robot manipulation, which in turn demands substantial GPU resources. For instance, both OpenVLA [1] and SpatialVLA [2] are trained using 64 NVIDIA A100 GPUs. In contrast, our model achieves superior performance across both real-world and simulation experiments (as shown in Tables 1–6 of the main paper) while using fewer GPU resources. We train our model on AMD MI300 (192G) GPUs as AMD GPUs are the compute resources currently available to us. Nevertheless, our model is compatible with NVIDIA GPUs, such as the A100, and can be trained on them if such resources are available.
>
> **Q3: Visual Imagination Dependency: Unclear how much visual prediction helps over action-only baselines.**
>
> Please refer to our response to **Q1** for the comparison with baselines.
>
>
> **Q4: Skill Transfer Scope: Skill generalization tested on a limited set of tasks; broader testing would strengthen results.**
>
> We appreciate the reviewer’s suggestion. Due to the time constraints of the review cycle, we were unable to conduct a broader evaluation. However, we plan to expand our experiments to a wider range of tasks in future work to more thoroughly assess the model’s skill transfer capabilities.
>
>
> **Q5: Efficiency: Can RoboSeer skip visual imagination inference when only actions are needed (e.g., for low-latency execution)?**
>
> We thank the reviewer for the insightful question. Currently, our model cannot entirely skip visual imagination inference, as the vision latent tokens are designed to interact with the action tokens during inference, mirroring the joint prediction strategy used during training. However, we can skip the computation in the transformer's vision output layers and omit the video frame decoding process to reduce computation.
>
> As shown in Lines 17–18 of the supplementary material, our current model achieves a control frequency of approximately 3 Hz during real-world deployment on a single GPU. More recently, by integrating acceleration techniques including X-DiT and TEA-Cache and distributing inference across 4 GPUs, we have reduced latency to approximately 0.38 seconds—enabling a control frequency close to 8 Hz.
>
>
>
> **Q6: Failure Modes: What happens when visual imagination is incorrect? Is action prediction always degraded?**
>
> Our observations indicate that failed visual imagination often leads to incorrect action prediction. We analyzed 60 randomly sampled failure cases in which the imagined videos either failed to reflect the instruction intent, exhibited physically implausible changes, or appeared visually incoherent. Among these, 55 episodes (91.7\%) resulted in failed actual executions. This suggests that inaccurate visual imagination is a major contributor to incorrect action predictions.
>
> **Q7: Domain Transfer: Have you tried generalizing to new environments (e.g., different lighting or backgrounds)?**
>
> As noted in Lines 205–206 of the main paper, the SIMPLER (VA) benchmark [3] introduces substantial domain variations, including changes in background, camera viewpoint, lighting conditions, and table textures—collectively referred to as "environment variants" [3]. These variations differ significantly from the original training distribution. As shown in Table 1 of the main paper, our model achieves the top performance under the SIMPLER (VA) setting, demonstrating its robustness and strong generalization ability to novel environments.
>
> Furthermore, as illustrated in the third row of Figure 2 in the supplementary material, the SIMPLER (VA) setting includes previously unseen tables and backgrounds not present in the original Google Robot dataset. Despite these domain shifts, our model maintains high-quality visual imagination and preserves consistency between imagined and actual executions. These results indicate that our model enables effective domain transfer across diverse and challenging visual environments.
>
>
>
>
> **Q8: Release Plans: Will the model, code, and pre-trained weights be made available?**
>
> Yes, we will publicly release the model, code and pre-trained weights to the community.
>
>
>
> **Q9: Smaller Models: How does RoboSeer perform with smaller video generators (e.g., CogVideoX-B)?**
>
> We appreciate the reviewer’s suggestion on evaluating smaller video generation models. In our work, we adopt CogVideoX-5B as the backbone because it supports image-to-video generation, which is essential for our setting where the model predicts future visual imaginations (i.e., videos) and actions from a single observed image. We did not evaluate a smaller variant, such as CogVideoX-2B, because it only supports text-to-video generation and lacks the necessary image-to-video capability.
>
> Additionally, in Table 7 of the main paper, we reported the results using OpenSora-1.1 which has 1B parameters. The performance is significantly lower compared to our model based on CogVideoX-5B. But it is important to note that there are many differences between these two models, including architecture, training data, and other factors, and model size alone may not fully explain the performance gap. There's rapid progress in the field of video generation and we plan to explore more efficient and capable image-to-video foundation models to improve our method's inference speed while maintaining high manipulation performance.
>
>
> **Reference**
>
> [1] OpenVLA: An Open-Source Vision-Language-Action Model
>
> [2] SpatialVLA: Exploring Spatial Representations for Visual-Language-Action Model
>
> [3] Evaluating Real-World Robot Manipulation Policies in Simulation (SIMPLER)
>
>
> *Once again, thank you for your valuable time and effort in reviewing our work.*
>
> Authors of Paper 7746

---

> > ### Comment · Reviewer_4hpC · 2025-08-05
> >
> > Thank you to the authors for the detailed and thoughtful responses. I am satisfied with the clarification regarding the theoretical motivation for dual prediction and the additional ablation results. These additions strengthen both the conceptual and empirical contributions of the paper.
> >
> > I will maintain my original score.

---

> > > ### Author Response · Authors · 2025-08-05
> > >
> > > Thank you for the additional comments. We're glad to see that our clarification and additional ablation study have addressed your questions. We will revise our paper accordingly and incorporate the added studies. Thank you again for your review and insightful suggestions.
> > >
> > > Authors

---

### Official Review · Reviewer_kjNa · 2025-07-04

**Clarity:** 4
**Significance:** 4
**Originality:** 3
**Rating:** 4
**Confidence:** 4

**Summary:**

RoboSeer introduces a vision-language-action model that, unlike prior work relying on pre-trained understanding backbones, leverages large-scale video diffusion models to perform dual prediction: it jointly forecasts the next sequence of 7-D robot actions and “imagines” the corresponding future video frames depicting the result of executing those actions. Built atop a multi-modal Diffusion Transformer initialized from CogVideoX, RoboSeer is conditioned on a language instruction and the current visual latent, and trained to denoise both future action chunks and frame latents simultaneously. Through in-domain and generalization experiments—in both SIMPLER simulation (WidowX and Google robots) and real-world with a Realman arm—RoboSeer not only achieves state-of-the-art success rates but also reveals a strong quantitative correlation between the fidelity of its visual imaginations and actual task success, suggesting that high-quality video prediction serves as an implicit reliability measure for action plans.

**Questions:**

* Action Conditioning: Is there a causal mask between future video latents and actions—i.e., can actions steer video prediction?

* Causal Decoupling: Have you considered ablating or “masking” the video-prediction loss (e.g., training without the visual imagination objective) to causally isolate its effect on action accuracy? A controlled ablation would clarify whether visual generation truly drives performance improvements.

* Inference Efficiency: What is the end-to-end inference latency on the Realman robot when sampling future frames and actions? If the model requires tens or hundreds of milliseconds per diffusion step, this could limit real-world responsiveness—please report wall-clock timings and discuss potential speed-accuracy trade-offs.

* Video-Action Correlation Analysis: You demonstrate a strong empirical correlation between imagined and actual trajectories. Can you provide insights into why this holds? For instance, is it driven by dataset coverage (e.g., similar scenes during training) or by model generalization?

* Generalization Alignment: Does the generalization behavior of video prediction mirror that of actions? In other words, when encountering novel objects or skills, is one modality easier to generalize than the other? Please consider separate generalization metrics for each modality to better understand their interplay.

**Ethical Concerns:**

["NO or VERY MINOR ethics concerns only"]

**Final Justification:**

I will increase my score to accept.

**Limitations:**

Please discuss the latency of video diffusion-based inference on real robots.

**Paper Formatting Concerns:**

No.

**Quality:**

3

**Strengths And Weaknesses:**

Quality: The paper presents a rigorously evaluated approach, with clear experimental protocols covering in-domain tasks and two types of generalization (novel objects and cross-embodiment skills) across both simulation and real robots.

Clarity: The architecture and training objectives are described in a straightforward manner, with helpful figures illustrating both the model design and correlation analysis. However, some implementation details—such as the real-time inference latency on physical hardware—are omitted, which could affect reproducibility in real-world settings.

Significance: By shifting from perception-only to generation-augmented planning, RoboSeer leverages advances in video diffusion to unlock stronger generalization. The demonstrated video-action correlation insight is particularly impactful, suggesting a built-in self-evaluation mechanism.

Originality: Adapting a large pre-trained video generator for joint action and visual outcome prediction in robotics is relatively new, setting RoboSeer apart from concurrent VLA models that mostly focus on understanding. The dual-modality denoising approach represents an interesting fusion of diffusion modeling of actions and future forecasting.

Weaknesses: The reliance on heavy diffusion sampling (50 denoising steps) raises questions about run-time efficiency in real applications. The paper does not explore whether simpler or faster generative backbones could achieve comparable performance. Additionally, while correlation is shown, the causal relationship between imagination quality and action accuracy remains unverified (e.g., through causal masking or ablation).

---

> ### Author Rebuttal · Authors · 2025-07-31
>
> Dear Reviewer kjNa,
>
> We appreciate your thoughtful feedback to our work. Please find our responses to your questions below.
>
>
> **Q1: Action Conditioning: Is there a causal mask between future video latents and actions—i.e., can actions steer video prediction?**
>
> Thank you for the insightful question. In our current implementation, video latent tokens and action tokens are fully bidirectionally visible to each other; we do not apply a causal mask between them. This design choice is based on the fact that video and action sequences are temporally synchronized but belong to different modalities. Allowing mutual visibility enables each modality to inform and guide the prediction of the other. This setup is analogous to joint video-audio generation, where synchronized but distinct modalities benefit from cross-modal attention to improve overall generation quality.
>
>
> To further examine whether a causal masking strategy offers advantages over our default bidirectional approach, we conducted an ablation study in which a causal mask was applied: action tokens could attend to video tokens, but video tokens could not attend to action tokens. This experiment was conducted in the SIMPLER (VM) environment, following the same setup as Table 1 in the main paper. The results are shown below:
>
> | Method | Pick up Coke Can | Move Near |Open/Close Drawer | Average |
> |:------|:--------:|:-------:|:-------:|:-------:|
> | Default |   92.3      |     82.9  |     66.2  | 80.4 |
> | Causal mask|   89.3    |     76.2  |    61.1 | 75.5 |
>
> We observed a consistent drop in performance under the causal masking strategy. This indicates that allowing video tokens to access action tokens enables the model to better capture how specific actions drive physical changes in the visual scene. In other words, the bidirectional interaction between action and vision modalities enhances coherence and alignment—allowing actions to more effectively guide video prediction, while video prediction, in turn, supports more accurate action prediction.
>
> We will include this analysis in the revised paper.
>
>
> **Q2: Causal Decoupling: Have you considered masking the video-prediction loss (e.g., training without the visual imagination objective) to causally isolate its effect on action accuracy?**
>
> The following table presents the performance of our variants with and without the video prediction loss, evaluated in the SIMPLER (Visual Matching) environment. When the video prediction loss is removed, the average performance drops significantly from 80.4 to 27.0, highlighting the effectiveness of our default strategy, which jointly predicts visual imagination and actions.
>
> | Video Loss | Pick up Coke Can | Move Near |Open/Close Drawer | Average |
> |:------|:--------:|:-------:|:-------:|:-------:|
> | Without |   35.6      |     29.1  |     16.2 | 27.0|
> | With  |   92.3      |     82.9  |     66.2  | 80.4 |
>
> We will include this study in the revised version.
>
>
> **Q3: Inference Efficiency: What is the end-to-end inference latency on the Realman robot when sampling future frames and actions?**
>
> As shown in Lines 17–18 of the supplementary material, during real-world deployment, we predict 13 frames using 10 denoising steps, resulting in approximately 1.1 seconds to generate 6 action tokens (with the first 3 used for execution), corresponding to an effective control frequency of ~3 Hz.
>
> To further improve efficiency, we integrate video generation acceleration techniques such as X-DiT and TEA-Cache. With inference distributed across 4 Nvidia A100 GPUs, the latency is reduced to approximately 0.38 seconds for generating  6 action tokens (with the first 3 used for execution), achieving a control frequency of ~8 Hz.
>
> We acknowledge the trade-off between performance and accuracy. Training a large, high-performance video generator from scratch would require thousands of GPUs; therefore, we initialize our model with publicly available pre-trained video generators, whose architecture cannot be easily modified. In future work, we plan to explore distillation techniques to compress large, general-purpose video generators into compact, robot-specific ones, thereby improving inference efficiency.
>
>
>
> **Q4: Video-Action Correlation Analysis: You demonstrate a strong empirical correlation between imagined and actual trajectories. Can you provide insights into why this holds? For instance, is it driven by dataset coverage (e.g., similar scenes during training) or by model generalization?**
>
> First, our training data consists of temporally aligned robot manipulation videos and their corresponding action sequences. Our model learns the cross-modal correspondence—particularly temporal alignment—between the visual and action modalities, enabling it to jointly predict both. Each modality provides complementary information to the other.
>
> Second, we leverage large-scale pre-trained video generators with strong generalization capabilities in video generation—i.e., the ability to generate novel content during inference. Our model can be viewed as a form of post-training on top of these video generators, allowing it to inherit their generalization capacity. As a result, the model can imagine plausible visual executions for novel environments, skills, and objects. Since it has already learned the alignment between imagined visuals and corresponding actions, this generalization ability transfers to the action domain, leading to improved generalization in robot manipulation tasks.
>
>
> **Q5: Generalization Alignment: Does the generalization behavior of video prediction mirror that of actions? In other words, when encountering novel objects or skills, is one modality easier to generalize than the other? Please consider separate generalization metrics for each modality to better understand their interplay.**
>
> We conduct an analysis in the SIMPLER environment to examine the generalization behavior of video prediction and action prediction on novel objects and unseen skills. The experimental setup follows the same configuration as Tables 2 and 3 in the main paper.
>
> The table below reports the success rates for both actual executions and visual imaginations. For actual execution, we can directly determine whether each task succeeds or fails and compute the success rate accordingly. In contrast, visual imagination produces video outputs, which cannot be automatically evaluated for success. Therefore, we manually inspect each imagined video to assess whether the task was successfully executed (a record is viewed successful if the visuals well follows the given instruction and there are no noticeable geometric distortions of objects or violations of physical plausibility).
>
> We observe that most visual imaginations result in reasonable execution outcomes—for example, achieving an 84.0\% success rate on novel objects and a 63.4\% success rate on new skills. In comparison, the actual execution success rates are lower. This discrepancy is expected because real-world execution requires precise grounding of the robot in the physical environment, which introduces additional challenges. We also observed that in these generalization tests, a higher visual imagination success rate corresponds to a higher action execution success rate (84.0\%$\leftrightarrow$65.2\%, vs 63.4\%$\leftrightarrow$48.6\%).
>
>
>
> | Metric | Novel Objects | New Skills |
> |:------|:--------:|:-------:|
> | Visual Imagination Success Rate (human judge) |   84.0      |    63.4  |
> | Actual Execution Success Rate  |   65.2      |     48.6  |
>
> We will include this analysis in the revised version.
>
>
>
> *Once again, thank you for your valuable time and effort in reviewing our work.*
>
> Authors of Paper 7746

---

> > ### Comment · Reviewer_kjNa · 2025-08-07
> >
> > Thanks for the detailed response — the additional experiments are quite helpful. I will increase my score to accept.

---

> > > ### Author Response · Authors · 2025-08-07
> > >
> > > Thank you for your additional comments.  We're pleased that our responses and the additional ablation study have addressed your questions.  We will update our manuscript accordingly and include the new studies.  Thanks again for your professional review and valuable suggestions.
> > >
> > > Authors

---

### Comment · Area_Chair_A7uE · 2025-08-02
**Please Engage in the Discussion Period ASAP**

Dear reviewers,

This is your AC. The authors have provided rebuttals to your reviews. Please read them as soon as possible if you haven't already done so.
I would like to encourage you to **post further questions and/or comments** and engage in an open exchange with the authors. Thanks!

Best,
Your AC

---

### Comment · Area_Chair_A7uE · 2025-08-05
**New Rules from the Program Chairs: Reviewers will need to post comments before acknowledgement!**

Dear reviewers,

Thank you for your initial reviews and now the program chairs have issued new rules to encourage participation in this period. Note that the author-reviewer discussion period has been extended by 48 hours.

Under the new rules, you will need to post at least a comment before submitting the “Mandatory Acknowledgement”. Otherwise, you will be **flagged for “Insufficient Review”**.

Here are some general guidelines for the author-reviewer discussion period:
+ It is not OK to stay quiet.
+ It is not OK to leave discussions till the last moment.
+ If authors have resolved your (rebuttal) questions, do tell them so.
+ If authors have not resolved your (rebuttal) questions, do tell them so too.

Thank you for your efforts!

Regards,

-Your AC

---

### Public Comment · ~Xiuyu_Yang1 · 2025-12-25
**Code and Model Resources Release**

This is an excellent piece of work. We would like to ask whether the authors plan to open-source the code and models. At present, the links to the codebase and Hugging Face repository on the officially released homepage appear to be invalid. Thank you.

---

### Decision · Program_Chairs · 2025-09-17

**Decision:**

Accept (poster)

**Comment:**

## Summary
This paper introduces RoboSeer, a novel vision-language-action (VLA) model that leverages pre-trained video generation models to jointly predict future actions and their visual outcomes for robot manipulation. The approach represents a paradigm shift from "understanding" to "generation" in robot learning. All four reviewers lean towards acceptance, and after a thorough discussion period, their initial concerns were largely addressed by the authors' detailed rebuttal and additional experiments.

## Strengths identified by the reviewers
- Novelty and Paradigm Shift: The core idea of using a generative model to imagine future visual outcomes alongside predicting actions was seen as highly original and a significant departure from existing VLA models that focus primarily on perception (Reviewers kjNa, 4hpC).

- Strong Results: The paper presents comprehensive experiments across simulation and the real world, demonstrating state-of-the-art performance and impressive generalization to novel objects, unseen skills, and new environments (Reviewers 4hpC, Knht).

- Insightful Analysis: The demonstrated correlation between the quality of the "imagined" future video and the actual task success rate was highlighted as a particularly impactful finding, suggesting a built-in mechanism for self-evaluation (Reviewer kjNa).

## Weaknesses and Rebuttal

- Causality and Importance of Visual Generation: Reviewers kjNa and Knht questioned the actual contribution of the visual generation component to the model's performance. They requested ablations to isolate the effect of joint modeling and pre-trained weights.

  - Rebuttal: The authors provided extensive new ablation studies. They showed that removing the video prediction loss or training from scratch without pre-trained weights led to a massive drop in performance. Reviewers kjNa and Knht were satisfied with these results.

- Missing Related Work: Reviewer cgvg raised a major concern about the paper's claim to be the "first" in this area, pointing out a significant body of existing work using video generation for manipulation that was not cited or compared against.

  - Rebuttal: The authors acknowledged this oversight. They provided a detailed discussion of the papers mentioned by the reviewer. They committed to adding this discussion to the related works section and toning down their claims of novelty. Reviewer cgvg was satisfied with the authors' commitment to revise the paper, stating they would raise their score assuming the changes are incorporated.

- Inference Efficiency and Latency: Reviewer kjNa raised concerns about the real-world applicability given the computational cost of diffusion models.

  - Rebuttal: The authors provided concrete latency figures for their real-world deployment (~3 Hz control frequency), along with details on how they improved this to ~8 Hz using acceleration techniques. They acknowledged the trade-off and discussed future work on model distillation. This was part of the detailed response that led Reviewer kjNa to raise their score.

- Generalization to New Environments: Reviewer Knht suggested evaluating generalization to unseen environments.

  - Rebuttal: The authors clarified that the SIMPLER (VA) benchmark used in their experiments already includes significant domain variations (lighting, backgrounds, textures), and their model achieved top performance in this setting, demonstrating this capability. Reviewer Knht was satisfied with the authors' response and maintained their positive rating.

## Recommendation Justification
This is a strong paper with a novel and impactful idea. The core contribution—leveraging large-scale video generators for joint action and outcome prediction—is a significant step forward for robot manipulation. The authors provide compelling empirical evidence of their method's effectiveness and strong generalization capabilities. The initial weaknesses, particularly the missing related work and the need for more rigorous ablations, were thoroughly addressed by the authors in their rebuttal. The reviewers were satisfied with the responses, and the consensus is a clear acceptance. The authors have committed to incorporating the additional related work, discussions and experiments into the final version, which will further strengthen the paper. I recommend accepting this paper.